# Infodemic and Fake News Turning Shift for Media: Distrust among University Students

**Ana Pérez-Escoda** 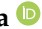

Department of Communication, Faculty of Communication and Arts, University Antonio de Nebrija, Sta. Cruz de Marcenado, 27, Madrid 28015, Spain; aperezes@nebrija.es

**Abstract:** In many parts of the world, long before social media, trust in media and journalism was fragile and shaky. Today, however, with an unprecedented information abundance, the situation has worsened because, in the high-speed information free-for-all of social media platforms and the internet, anyone can consume and produce. As a result, citizens find it difficult to discern what is real and what is fake. In this context, the aim of the study is to explore how information and fake news consumption affects the perception of media in terms of trust. The methodology applied for this purpose was a mixed method using both quantitative and qualitative data in order to provide not only descriptive data but more thorough results. For the quantitative analysis, a sample of 849 university students participated: from these, a smaller sample of 100 participated in the qualitative phase. Conclusions indicate that the distribution of fake news is worryingly associated with the media and, consequently, a concerning distrust of media is shown among participants who express feeling insecure, vulnerable, confused, and distrusting of media.

**Keywords:** information; media; social networks; fake news; disinformation; media distrust; media literacy

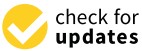



## 1. Introduction

The proliferation of the SARS-CoV-19 virus has infected the world's population since its appearance in early 2020, not only with the biological disease known as coronavirus, but also with another sociological and communicative pathology: disinformation, caused by an over-abundance of information, named as an "Infodemic" [1,2]. Thus, what was basically a theoretical concept in the field of journalism has become an endemic problem that concerns all social agents, from citizens to politicians and national and international institutions. The concerning dimension reached by this phenomenon requires concrete studies to obtain significant evidence to contain the global defenselessness of citizens, whereby fake news is becoming a threat not only to citizenship but also to democracies [3–5]. In this context, media outlets, which have traditionally been recognized as legitimate social agents to inform society, have been affected by this situation, compromising their role [6]. The problem alters democratic systems through a flow of truthful and credible information that provides citizens with the possibility of making informed decisions. Without this possibility, the amount of information and an increase in fake news become a turning point for media outlets in terms of trust and reliability, especially among young people who are mostly considered digital natives and are highly influenced by their digital consumption and interaction [7].

In this sense, the purpose of this work is to provide not only a descriptive picture but also a deeper one on how information and fake news consumption affects the perception of media in terms of trust. Through an exploratory and ethnographic approach (which allowed a systematic approach to a concrete reality since, based on this knowledge, actions were proposed), possible strategies to be developed through the promotion of digital literacy in social networks could be defined in order to counteract a situation that not only lasts over time, but seems to have come to stay.

## 1.1. Disinformation, Fake News and Social Networks

Although disinformation has multiple origins, since the appearance of the COVID-19 virus, its proliferation has been widely linked to social networks and instant messaging applications that, through smartphones, easily and quickly penetrate the population that consumes them, mostly through social networks, as has been shown in several previous studies [8,9]. As highlighted by Sivasankari and Vadivu [10], people mainly rely on social media for any form of information sharing and also collect information through social media platforms to be informed. In line with this argument, the work published by Shrestha and Spezzano [11] highlighted the rapid spread of fake news in social networks and the use of social media for daily news.

Disinformation, defined as "intentional dissemination of non-rigorous information that seeks to undermine public trust, distort facts, convey a certain way of perceiving reality and exploit vulnerabilities with the aim of destabilizing" [12] (p. 2), is associated with the term fake news. Tandoc and others [3] conducted an in-depth study on this term based on the momentum received in recent political issues and, more recently, in social and health issues. In their study, they point out that the term is not new. The authors define a typology or categorization that may shed light on the root of disinformation: (1) news satire, (2) news parody, (3) fabrication, (4) manipulation, (5) advertising, and (6) propaganda [3] (p. 147). With this categorization, it could be inferred that a verification work by citizens is not feasible, as there is no pattern that describes them and can be easily detected.

In the closest international context, the European Union has made continuous efforts to counteract the phenomenon through actions such as the Eurobarometer against fake news and online disinformation, or the directives that, since 2013 (Directive 2010/13/EU), have attempted to establish regulatory frameworks with active proposals to combat fake news: groups of specialists, the creation of fact-checkers, or the use of artificial intelligence. However, these actions have not been sufficiently effective in the face of the virulence of the infodemic, as recognized by the World Health Organization (WHO) [2].

The challenge presented by misinformation must be addressed as a global issue inherent to all fields and not as a new one, as noted in the report "Bias, Bullshit and Lies: Audience Perspectives on Low Trust in the Media" [13], except one that is more dangerous than ever before. For a better understanding of the framework, related to the term "infodemic"—understood as the problematic amount of information to be managed by citizens—, there are other issues affecting the situation. From a sociological and political point of view, populism arises with a far-reaching and never seen before influence, as demonstrated in several previous works such as in Alonso-Muñoz and Casero-Ripollés' [14] analysis of discourses on Twitter, Keyes' [15] study of post-truth as a lie (falsehood), and Masip and others' [16] focusing on polarization, filter bubbles, and echo-chambers which nurture disinformation.

## 1.2. Media and Audiences

In this situation, it is widely accepted that hyperconnectivity and the global access to the media through social media have increased the ease of the dissemination of fake news, directly affecting relations between media and audiences [4]. As previously studied by Williams [17], the relationship between an individual's levels of media trust and news attention can be observed from different levels of trust, with especial emphasis given to internet news. In this regard, studies following Williams' focused on trust in traditional media, linking an audience's trust in institutions with media trust [18], for instance, or by exploring the association between mainstream news and exposure to online news [19]. However, social media as containing specific digital scenarios, considered as sources of information, need to be explored in terms of trust and media, taking into account the current situation in which traditional media outlets use social networks in a double way: firstly, to spread their agenda as an alternative way to engage with audiences, and, secondly, to adopt an emerged role of "attention drivers", as stated by Ren and others [20]. The Digital News Report 2021, published by the Reuters Institute for the Study of Journalism [21], noted

that 58% of their survey population, which included 40 countries, was concerned about what is false and real on the internet in different areas (health–COVID-19, 54%; politics, 43%; celebrities, 29%; climate change, 20%); however, and despite admitting their lack of trust, a large part of the population continued to use social networks as their main source of information. Recent studies regarding this issue shows this paradox among audiences, especially for the youngest—although they do not trust in social media, they declare using social media to be informed [7,17,18].

This context places a greater responsibility on media and journalists, whose work is exposed to a higher expectation of veracity and credibility [22]. Not only professional ethics but also the trust of their audiences is damaged by an alarming context in which social networks appear as major competitors to traditional media, forcing audiences and the media to follow the flow of information online despite the danger of being compromised by the intervention of actors and users who are not legitimized to inform [23,24]. In this sense, although media outlets have tried using innovative efforts through digital environments and new formats [25,26], the abundance of information and the paradigm shift in communication still shows that there is more to be done to counteract the negative effects evident in the prosumer era.

The evolution must be directed towards citizenship education in order to achieve a balance in society. Social media as a major promoter of engagement in society [27] and industry [28] must be included as part of formal education and as real scenarios in which citizens need to manage efficiently. New skills need to be taught in all levels of education for them to become good practices for citizens [29,30].

It is important to mention this in the context of the growth of social networks around the world in recent years, and has been especially significant during the pandemic, wherein social media acted as a window of relief from the global situation of lockdown. This has led to the extensive use of these virtual spaces by all population groups, though with particular importance for the youngest or those in their formal learning years [31].

Using the aforementioned framework, the study aims to analyze how information and fake news' consumption affects an audience's perception of media in terms of trust. For this purpose, there were three specific objectives: firstly, to analyze the audience sample's use of social networks in terms of time, specific use, and role adopted; secondly, to examine fake news' reception and the perception related to media outlets; thirdly, to study the perception of trust and its relationship with sample's age. In light with these main objectives, subsequent research questions emerged:

RQ1: Is there any correlation between the age and the time spent on social networks?

RQ2: Is there any significant correlation between age and the use of different social networks?

RQ3: Is fake news perceived as a positive or negative phenomenon?

RQ4: Are media outlets perceived as distributors of fake news?

RQ5: Does trust depend on the age of the audience?

## 2. Materials and Methods

In order to achieve the proposed objectives, the research was articulated from a dual methodological approach to offer a deeper perspective of the results [32]. Both quantitative analysis and qualitative descriptions based on semi-structured interviews were used in order to thoroughly examine the obtained results. This method seeks to find solutions or propose alternatives to situations detected from this double process.

### 2.1. Quantitative Research

The quantitative research was based on the questionnaire, as it was considered to be the most accurate method for collecting data for an exploratory and descriptive study [33] and would allow us to obtain a foundation. The instrument was adapted from a previous questionnaire, used and validated in [7], to focus on the objectives. The questionnaire was adapted with four blocks of items: (1) sociodemographic variables: age, gender,

country, autonomous community, current degree; (2) social media consumption; (3) fake news perceptions; (4) trust in social agents. Each block was formed as a study construct, integrating a certain number of items, as shown in Table 1.

**Table 1.** Definition of the constructs under study and number of items studied.

| Study Concepts | Conceptual Definition | Items | Cronbach's Alpha |
|---|---|---|---|
| Social media consumption | A new shift paradigm in communication field is assumed in which young people mostly consume information and interact in a digital environment fostered by social media [34]. | 9 | 0.78 |
| Fake news reception and perception | Based on the perception of consuming and receiving misleading or incorrect information that is passed off as real information [35]. | 21 | 0.86 |
| Trust | It is considered as a subjective and relational concept, built on experience or, in its absence, on the expectation that the interaction with the trustee (media) would lead to gains for the trustor (audience) [36] (p. 4). | 7 | 0.71 |

Own elaboration.

In the second stage, after the design of the questionnaire, the instrument was sent to a group of experts (N = 5) to validate the clarity and relevance of the study constructs. Their evaluation recommended a reduction in items (from 95 to 89), namely, in those that were redundant or inappropriate. To conclude this stage, the reliability and consistency of the instrument was studied by applying Cronbach's alpha with results that ranged between 0.71, 0.78 and 0.86, respectively, for the mean of the variables of each of the three constructs (as shown in Table 1). According to Tavakol and Dennick [37], obtained values ≥0.7 were acceptable for a proper consistency of the instrument. Most of the variables were characterized as ordinal qualitative variables with a Likert-type scale with five values, where "1" meant "not at all or never" and "5" meant "a lot or all", and nominal qualitative variables, where the numerical values assigned are not scaled but are numerically coded.

As for the sample under study, the chosen sampling technique was non-probabilistic and included convenience sampling, in which the sample is available at the time and during the period of the research, as Vilches [33] pointed out. A total of 849 subjects (M = 26.8; SD = 8.24; Min. = 18; Max. = 51) from eight different degrees—six undergraduates, representing 40.2% (N = 341), and two postgraduates, representing 59.8% of the sample (N = 508)—were recruited from two different faculties: Communication and Education. The gender distribution was 66.8% female (N = 567) and 33.2% male (N = 282).

*2.2. Qualitative Research*

The quantitative research was based on an ethnographic perspective as the methodology aims to describe the qualities of a phenomenon (in this case, the sample's perception) before answering questions related to social media consumption, fake news reception, and the perception and trust towards media. Therefore, for the analysis of the qualitative data, a content analysis was carried out in which the stages of the analytical process of the data were followed based on the scheme of Miles and others [38]: (1) data reduction, with the separation of units, identification, and classification of elements, and grouping; (2) arrangement and transformation of the data; (3) obtaining results and verification of conclusions. This analysis was carried out on a single item that collected textual information pertaining to their coding and categorization. In this case, only 100 students participated in this phase. For the first stage of data reduction, a frequency analysis of the most frequently used words was carried out in order to explore the most relevant concepts in the results obtained, resulting in a conceptual map of categories being derived from the qualitative analysis results. The list was purged of empty words and the most representative words were numerous, as shown later in the results.

The analysis of the results was carried out with two different tools: the quantitative results were transferred to the SPSS statistical program (version 25) and the qualitative results were transferred to the NVIVO 10 program, transforming the textual data for interpretation into structured data. Ethical approval was not needed in this study as it only collected perception; however, personal informed approvement was provided by every individual after answering the questionnaire and after participating in the interviews.

## 3. Results

### 3.1. Results for the Quantitative Analysis

The quantitative analysis of the results was carried out in accordance with the research objectives; it is important to note that, given the volume of data obtained, only some of them are shown. Firstly, normality tests were carried out on the distribution of the sample, and the Kolmogorov–Smirnov test showed that there is no normality ($p < 0.05$), which determined the non-parametric nature of the data. In this regard, the correlation coefficient of Spearman was applied to observe the correlation among variables.

#### 3.1.1. Social Media Use and Engagement

In the first section of the study, we addressed RQ1: Is there any correlation between the age and the time spent on social networks? Spearman's correlation coefficient showed a moderate negative correlation (R = −0.516; $p = 0.00$), but one that was statistically significant ($p < 0.05$). Although the result that the older the age, the less time spent on social networks is to be expected, it is nevertheless noteworthy that this correlation is moderate, which reinforces the thesis that social networks are attracting all population niches regardless of their age. This accentuates the importance of considering network consumption data to make any kind of diagnosis when talking about information consumption and fake news.

This first result is clearly observed in Figure 1, in which bigger differences are appreciated in less hours spent using social networks and where the difference between undergraduate and postgraduate students is significant (note the percentages taken by time slot).

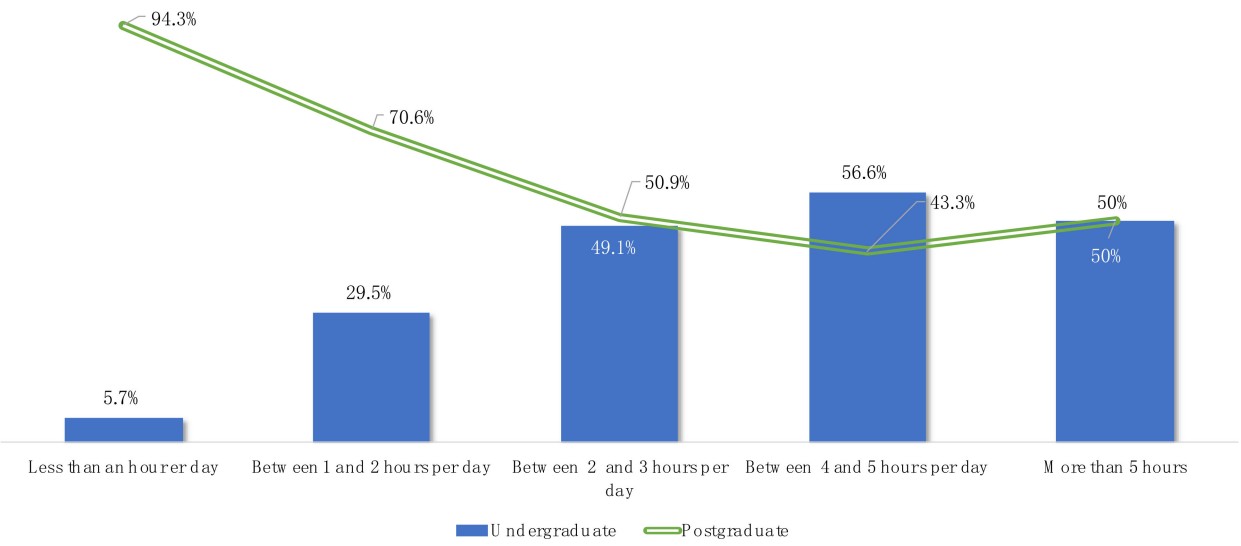

**Figure 1.** Results for differences in time spent consuming social networks.

Main descriptive data are shown in Table 2 regarding the frequency in the use of different social networks. According to the obtained results, it seems clear that WhatsApp is the most used, as more than a half of the respondents reportedly use it "Always" (56.2%) and 81.2% use it "Frequently and Always". Results showed significant correlations in all cases ($p < 0.05$) of coefficient correlations and chi square tests for the variables "age" and each social network studied.

**Table 2.** Statistical descriptive results for the social networks' consumption.

| Social Network | M | SD | Never | Rarely | Sometimes | Frequently | Always | N |
|---|---|---|---|---|---|---|---|---|
| Facebook | 1.89 | 1.144 | 50.6% | 26.5% | 10.8% | 7.5% | 4.5% | 849 |
| Instagram | 3.63 | 1.387 | 13.0% | 9.3% | 15.9% | 25.6% | 36.3% | 849 |
| TikTok | 1.92 | 1.31 | 58.7% | 14.6% | 9.4% | 10.4% | 6.9% | 849 |
| Twitter | 2.47 | 1.415 | 33.5% | 26.7% | 12.4% | 14.1% | 13.3% | 849 |
| WhatsApp | 4.31 | 0.913 | 0.2% | 5.4% | 13.2% | 25.0% | 56.2% | 849 |
| YouTube | 3.12 | 1.103 | 4.2% | 29.6% | 28.9% | 24.3% | 13.1% | 849 |

Own elaboration.

The last variable studied in this first construct relays the engagement of each participant when using or consuming social networks as seen in Figure 2. In this case, five different levels of participations were established in order to measure how engaged the participant is when using social networks. The first stage, corresponding to the lowest, was "Consumption without participation"; this stage corresponded to the concept of lurkers, according to Edelmann [23]. The highest was defined as "Sharing, commenting, mentioning and creating debate".

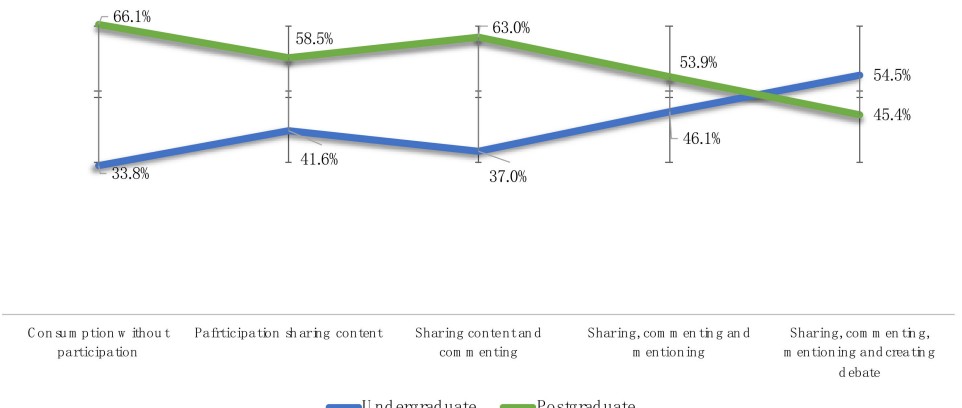

**Figure 2.** Level of engagement in social networks in percentages.

It is worth mentioning that, over and above the expected age difference presented in the first levels, the results showed barely any difference in the last two levels of engagement, where the age difference seems to be diluted; namely, "Sharing, commenting and mentioning" was higher among postgraduate students (53.9%) and "Sharing, commenting, mentioning and creating debate" had a percentage of 54.5% for undergraduate and 45.4% for postgraduate students.

3.1.2. Fake News Perceptions

Fake news perception was studied with three different variables to answer RQ3: Is fake news perceived as a positive or negative phenomenon?; RQ4: Are media outlets perceived as distributors of fake news?

Having in mind that fake news could include different formats and types, the feeling related to fake news was measured with four different variables: humor, entertainment, distrust, or manipulation. The results obtained show no statistically significant differences ($p < 0.05$) in the case of perceiving fake news as humor ($p = 0.915$), entertainment ($p = 0.172$), or manipulation ($p = 0.109$). However, it seems interesting to highlight that, in the perception of fake news as "Distrust", results show statistically significant differences ($p = 0.000$). This can be observed in Figure 3, where different slot ages are shown in percentages. Surprisingly, it is younger students who associate fake news with distrust.

PERCEIVED AS DISTRUST

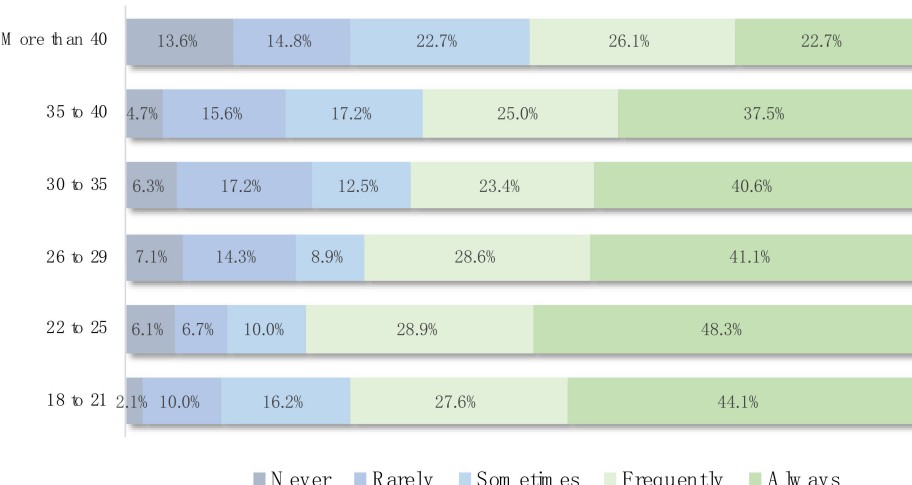

**Figure 3.** Percentages showing fake news perceived as distrust in slot ages of the participants.

The distribution of fake news associated with media outlets was perceived for the total of the sample, as shown in Figure 4 (M = 2.72; SD = 1.037). General data show that a concerning 50% of the sample associated fake news with media outlets in different levels (Sometimes, 25.7%; Frequently, 19%, and Always, 5.4%).

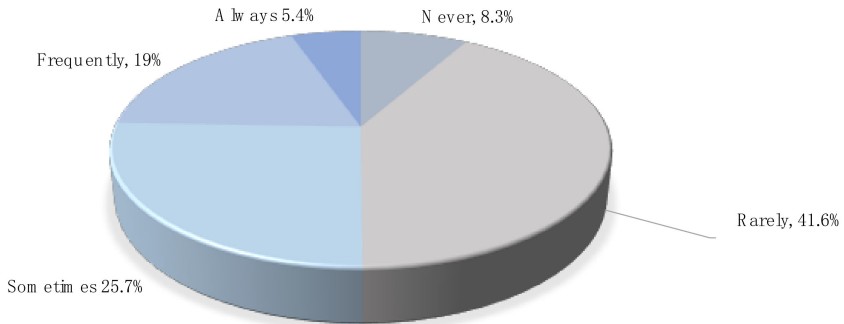

**Figure 4.** Percentage of participants perceiving fake news associated with media outlets.

As for correlation with age, the chi square test conducted shows statistically significant differences ($p < 0.05$) that can be appreciated in Tables 3 and 4, showing percentages per column and row, respectively. Undergraduate students (Slot 18 to 21) show the highest level of perceiving media outlets as distributors of fake news: Moreover, 34.8% declare that this is "Always" the case; the same percentage is revealed for "Frequently", and 41.3% is shown for "Sometimes". The differences are clearly marked with respect to the older participants.

**Table 3.** Participants' perception of media outlets as distributors of fake news (per column).

| Slot Ages | Never | Rarely | Sometimes | Frequently | Always | N |
|---|---|---|---|---|---|---|
| 18 to 21 | 48.6% | 40.8% | 41.3% | 34.8% | 34.8% | 340 |
| 22 to 25 | 24.3% | 17% | 22% | 26.7% | 26.1% | 180 |
| 26 to 29 | 8.6% | 11.9% | 13.8% | 14.9% | 21.7% | 112 |
| 30 to 35 | 7.1% | 6.2% | 7.8% | 10.6% | 6.5% | 64 |
| 35 to 40 | 2.9% | 11.6% | 4.1% | 6.2% | 4.3% | 64 |
| More than 40 | 8.6% | 12.5% | 11% | 6.8% | 6.5% | 88 |
| TOTAL | 100% | 100% | 100% | 100% | 100% | 849 |

Own elaboration.

**Table 4.** Participants' perception of media outlets as distributors of fake news (per row).

| Slot Ages | Never | Rarely | Sometimes | Frequently | Always | N |
|---|---|---|---|---|---|---|
| 18 to 21 | 10% | 42.4% | 26.5% | 16.5% | 4.7% | 340 |
| 22 to 25 | 9.4% | 33.3% | 26.7% | 23.9% | 6.7% | 180 |
| 26 to 29 | 5.4% | 37.5% | 26.8% | 21.4% | 8.9% | 112 |
| 30 to 35 | 7.8% | 34.4% | 26.6% | 26.6% | 4,7% | 64 |
| 35 to 40 | 3.1% | 64.1% | 14,1% | 15.6% | 3.1% | 64 |
| More than 40 | 6.8% | 50% | 27.3% | 12.5% | 3.4% | 88 |
| Total | 8.3% | 41.6% | 25.7% | 19% | 5.4% | 100% |
| N | 70 | 353 | 218 | 161 | 46 | 849 |

Own elaboration.

### 3.1.3. Trust in Media Outlets and Social Actors

Data included in this paragraph show results to answer RQ5: Does trust depend on the age of the audience? As for the question related to trust in media, Figure 5 shows similar averages in all slot ages for Slightly (the second lowest level after Not at all): 31.8% (18 to 21), 37.8% (22 to 25), 35.7% (26 to 29), 29.7% (30 to 35), 46.9% (36 to 40), and 42% (More than 40). These data, along with the data obtained for the "Moderate" level, imply a very poor trust in media outlets. Focusing on data related to the highest levels of trust, the "Very" and "Extremely" results are as follows: 21.4% (18 to 21), 19.5% (22 to 25), 16.1% (26 to 29), 9.4% (30 to 35), 17.2% (36 to 40), and 13.6% (More than 40). Percentages in all age slots studied can be interpreted as low, and none of them reach 25%.

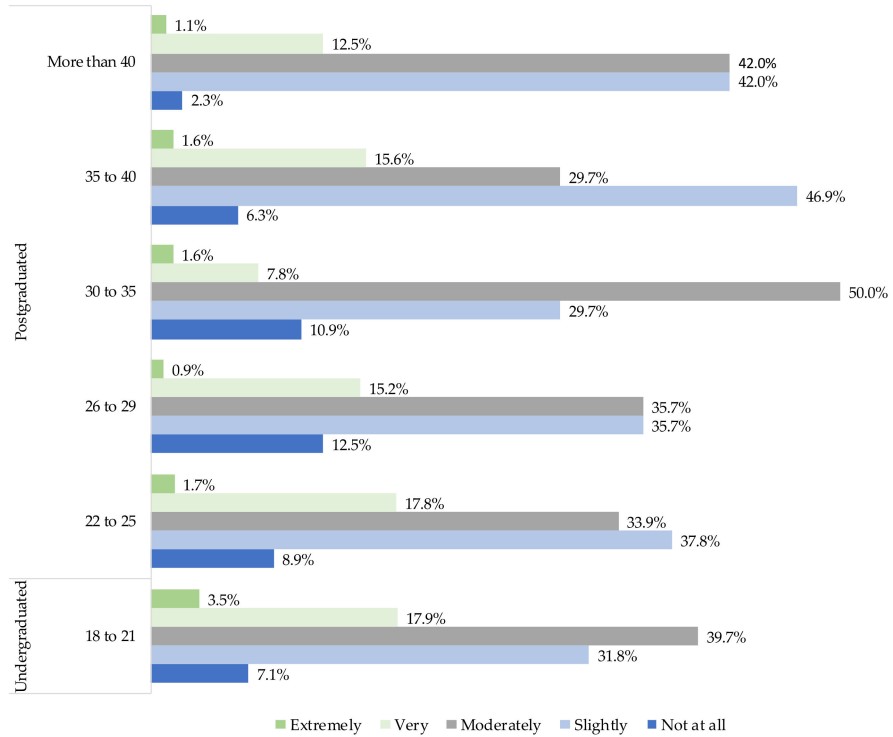

**Figure 5.** Percentages showing level of trust declared by participants in slot ages.

### 3.2. Results for Qualitative Analyses

From the content analysis of the qualitative data, it has been possible to recognize a series of concepts that represent the sample's perspective with categories [39], as shown in Figure 6, where discourses obtained from participants were categorized into main concepts.

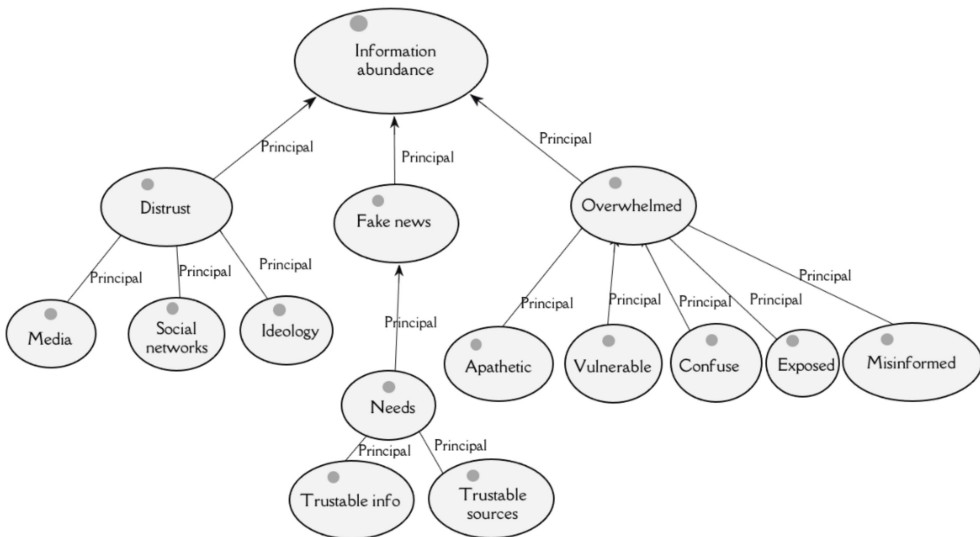

**Figure 6.** Conceptual map of categories. Own elaboration.

According to the results obtained from the first stage of the qualitative data analysis, a deeper analysis is presented below in two subsections. It is worth mentioning that each individual in the sample was coded with a P (person) and a number assigned from 001 to 100, as the total sample at this stage was 100 students.

### 3.2.1. Overwhelmed by a Tsunami of Information

One of the aspects that coincided in the majority of the sample analyzed in this phase is the overexposure to an avalanche of information that is very difficult to manage. It is interesting to mention the perceptions of overexposure, vulnerability, confusion, or insecurity shown by the participants. Several individuals mentioned "infoxication" that relates to the impossibility or difficulty of keeping informed or being able to choose trustable information.

P034: " . . . here comes a point where the volume of information is so vast and usually so loaded with ulterior motives, depending on the medium you are looking for, that I don't care if I spend a few days uninformed".

P045: "It is true that every time I want to inform myself about a particular news item or topic, what I do is to consult several news items related to that topic and extract my own opinion. Sometimes this gives me a feeling of insecurity, feeling "infoxicated" from the moment I start obtaining information until I am able to achieve what truly has happened".

P072: "Since the outbreak of COVID-19, we have been overloaded with constant news and information. I have become aware of the infoxication to which we are subjected on a daily basis, to the extent that this overexposure generates a sense of distrust and anxiety".

In this regard, the results show a clear link between the "amount of information" and "being misinformed". The paradox of a society with more access to information than ever before while being less informed than ever before is shown in the participants' statements:

P033: "Nowadays it benefits us in the way we can find any kind of information, it helps us to get along with our environment in a more global way and makes our lives a little easier. However, it is easy to fall into finding "what you want", in other words "what you want to read" for whatever reason, without checking the information, which makes us fall into the consumption of fake news. This leads to a society that is uninformed, but believes otherwise, and not only because of fake news but also because of the politicization of the media in many cases".

P091: "The over-information to which we are subjected has put in check the veracity of the content we receive, since we obtain it from so many media and in so many different ways that I believe we are beginning to be indifferent to its content".

P093: "I believe that we are surrounded by too much information and that the majority of the population is tired of so much false information".

3.2.2. Distrust, Fake News and Insecurity in Media

A second aspect widely underlined by the members of the sample is the distrust and insecurity generated by the media, although there is a clear tendency to blame the situation on social networks, as can be seen in some statements:

P024: "The survey has made me reflect on the low level of trust that I place in the media, it makes me reconsider the amount of fake news and hoaxes that are generated in order to discredit third parties and the reach they have thanks to the media and especially social networks. We live surrounded by this information and it is necessary to be critical, responsible and verify the information, otherwise we contribute to spreading these lies. However, as a result of this proposal to verify information, a doubt arises in my mind: who is right? How do I know if the information I am verifying in another media is real or just another fake new? Can we not trust anyone?".

P071: "I think that through the media and networks, they inject us with all kinds of information, much of it is fake news, they have bombarded the networks with this type of programs to sell free training courses, false news related to COVID, or hundreds of job offers that do not exist, which they take advantage of to collect people's data, to send you advertising, to sell you courses, etc. It is something that I find degrading. All of this means that in the end, we distrust everything, and we no longer believe what is published, whatever the medium, we live in a constant state of distrust".

The lack of trustworthiness of the media brings to the table the legitimacy of other social actors who may be more trustworthy:

P087: "Information is hampered by fake news, which raises doubts as to whether it is true or not, so that we do not believe much of the news. An important example to highlight is this reality with the COVID pandemic, since some media say one thing and others say another, without practically never coinciding in what they inform us; it is much more likely that a hospital worker in your city or town can give more accurate information about what is happening than the news providers themselves, since the media manipulate the information at their convenience in order to capture the consumer's attention, a sad fact since we can feel deceived".

The lack of credibility in the media is recurrently associated with social networks; in some cases, the distinction is made in favor of the traditional media, which exceptionally (in the sample analyzed) continue to have credibility as social actors legitimized to inform:

P052: "Personally, I would like to point out that I consider myself a person who uses social networks and digital media very assiduously. However, I use them for leisure and entertainment purposes, since if I must look for some kind of information or news, I usually resort to traditional media, as I give them greater credibility and they transmit me greater confidence and security".

Media politization seems to be an element of distrust or an element of filter bubbles:

P033: "This leads to a society that is misinformed, but believes otherwise, and not only because of fake news but also because of the politicization of the media in many cases. For example: Reading a news item about the management of COVID in a newspaper of one political leaning and the same news item in another of the opposite ideology. They seem to be different things even though they are the same news. I will believe the one that is more in line with my ideology".

P061: "It is clear that we live in an age of adaptation to new ways of receiving information. This has several advantages and at the same time disadvantages and associated risks. We can keep ourselves informed on multiple subjects that previously required effort and specialized media that were difficult to access, all in real time. We can access a multitude of interesting points of view. However, we run the risk that our echo chambers will only repeat the opinions we are willing to hear and that malicious broadcasters will send out deliberately false information. The current design of social media itself encourages

such manipulations and it is likely that in the medium-term regulations will be needed to dampen these risks".

The perceived solution to this situation relates to training or education:

P055: "We must learn to identify this fake news, we must go far beyond what we see and hear, so it is in our hands not to stay with the first news we see but to reflect, analyze and compare it, that is, it is necessary to develop a series of cognitive skills that are acquired through the educational process".

## 4. Discussion

The findings reveal a high concordance between the quantitative and qualitative data, reinforcing the descriptive trends observed in the data through the participants' statements. Research into media trust over time provides academic evidence that this is an essential concept for democracies. This is the case of the study conducted in 44 countries and authored by Tsfati and Ariely [40]. If audiences trust the media, they consume them, giving them legitimacy and credibility, and thus feeling involved in the flow of communication that enables them to make informed decisions as citizens and active participants in democratic scenarios. In this case, media trust constitutes a fundamental condition, as stated by Schranz and others [41]. On the contrary, as has been demonstrated in previous research from Tsfati and Cappella [42] and Ladd [43], if the relation of trust breaks down, skepticism arises first and distrust later. The study presented in this paper converges with this conception of a broken trust by showing worrying signs of mistrust on the one hand (in all age slots studied, more than 40% of the sample) and an alarming lack of trust on the other (none of the age slots reach 5% in the category trusting extremely in media).

The distribution of fake news is worryingly associated with the media, with almost half of the sample surveyed associating it with "Sometimes" (25.7%), "Frequently" (19%), and "Always" (5.4%). This denotes a discrediting of the media that perhaps needs further reflection as "the biggest challenge facing journalism", as stated by Fink [44].

Age is shown to be a fundamental aspect when it comes to finding differences in the use of social networks, but not in the time spent on them, as previously argued in Newman and Fletcher [14] and Loosen and others [19]. However, regarding the perception of fake news, it is the younger population (undergraduates 72.7%, Frequently and Always) who mostly associate fake news with distrust—a percentage that decreases as the age of the sample increases (reaching 48% in the case of participants over 40). Consequently, when analyzing the perception of the media as distributors of fake news, it is the youngest participants who associate this distribution the most (almost 70%) with distrust. In this case, the current study offers an innovative perspective in comparison to other studies such as that by Masip and others [17], where elder participants showed higher rates of distrust than younger ones. Moreover, perception is an innovative perspective offered in this field, giving a subjective overview of the sample and enriching previous works focused on fake news characterization, such as that by Oliveira and others [8], or on trust in political and regulative institutions, such as that from Tejkalová and others [36], fake news' automatic detection analyzed by Ranjan and others [9], and fake news' propagation, as studied by Sivasankari and Vadivu [10].

## 5. Conclusions

Mixed methodologies allow to obtain not only descriptive results that could provide an initial picture of the problem, but a deeper perspective from the sample analyzed. Therefore, in this regard, this is an innovative study which offers significant results not only for future studies in the same field but for media outlets or journalists, helping them with empirical evidence in the challenge of gaining their audience's trust. Results could help strengthen media strategies in the innovative design of new products or by introducing innovations [45].

The main conclusions from the study presented emerge as an important topic concerning for media and as a social issue. As described by a participant in the qualitative

analysis: "However, it is a difficult and confusing time to stay informed". Now more than ever, the empirical approach presented offers us the perceived solution and a clear line of action from the academic sphere: media literacy as an effective response. In this sense, and assuming the limitations of the study presented (as it is not representative of the population), it is worth highlighting that it is a sample with a common feature: they are all university students. In this sense, age is not important, but rather, the need for training as a fundamental and defining aspect of the different population niches.

The specific training action proposed as a conclusion is in line with the work of Pérez-Escoda and others [30], in which the UNESCO initiative for the integration of media literacy and the use of social networks is taken as a reference. This initiative is underpinned by six key elements: (1) Rights, (2) Education, (3) Voice, (4) Intercultural dialogue, (5) Ethics, and (6) Wise click. The proposal assumes a massive use of networks by the population (especially the younger ones) and is in line with EU best practices against disinformation. The attraction of the project is that it seeks to naturalize the use of social networks from a perspective that combines responsibility, education, participation, critical thinking, and citizen communication. It is worth mentioning that the author of this paper is implementing a pilot test for a specific training proposal based in the UNESCO that is structured in four steps: (I) Developing a proper digital identity; (II) Nurturing the personal learning environment (PLE) and personal network learning (PNL); (III) Creating a conscious role of content curators, which implies selection, evaluation, and critical thinking; (IV) Creating digital content for an active and responsible civic participation. In future papers, the results will be exposure as well as perceptions of students in this regard. Although the research presented is clearly limited as the results are not representative of the total population, it offers empirical evidence from a quantitative and qualitive perspective focusing on an audience's perceptions in a wide range of ages. This study sheds light for future research aiming to thoroughly investigate audiences' perspectives while taking into account the fact that, as the social media penetration in the global population increases, the less differences among ages we find.

**Funding:** This research received no external funding.

**Data Availability Statement:** Not applicable.

**Acknowledgments:** Thanks to all students that kindly agreed to collaborate with their experiences and perceptions.

**Conflicts of Interest:** The authors declare no conflict of interest.

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
