# Peer review of "Infodemic and Fake News Turning Shift for Media: Distrust among University Students"

_information, doi:10.3390/info13110523_

Round 1

Reviewer 1 Report

According to the expert, the submitted manuscript does not correspond to the journal’s scientific direction: “Information Studies that shall advance global and collaborative studies in the sciences of information, information technology and information society as a field in its own right, elaborate common conceptual frameworks and implement them in practice to contribute to mastering the challenges of the information age”.

The following remarks were spotted in the manuscript.

1. Who are the authors of the manuscript?

2. In the Introduction, references to sources prove that social media as a significant promoter of engagement in society and industry must be included as part of formal education, as real scenarios in which citizens need to manage efficiently. New skills seem to be required to be taught at all levels of education to become exemplary practices for citizenship.

It has been proven that “This challenge requires a new curricular approach towards digital and media literacy that allows substantial progress in overcoming risks derived from media manipulation, inappropriate uses of the internet, inappropriate consumption of information, disinformation and infoxication and the spread of fake news”.

Moreover, considering the literature review, it is unclear what is proposed to be investigated.

Research Question 1: Is there any correlation between age and time spent on social networks?

RQ2: Is there any significant correlation between age and the use of different social networks?

RQ3: Are fake news perceived as a positive or negative phenomenon?

RQ4: Are media perceived as distributors of fake news?

RQ5: Is trust a value in danger? Which social actors are more authorized?

The expert does not understand the scientific contribution that the authors make public. The introduction should also provide a deeper structure (1.1, 1.2, etc.) of the manuscript.

3. Materials and Methods

The sentence (lines 157 and 158) is probably unfinished. In another case, a period should be added at the end of the sentence.

General impression.

1. According to the expert, the authors should clarify the goal and tasks of the research so that they explicitly emerge from the literature review as unsolved problems.

2. It is necessary to specify the proposed approaches for achieving the goal and solving the tasks in Materials and Methods.

3. The results should be presented as solutions to the set tasks.

4. In the Discussion, the authors should compare the results with the known ones more clearly.

5. Conclusion section should be extended using: 1) numerical results obtained in the paper; 2) limitations of the proposed method; 3) perspectives of future research.

Author Response

All suggestions are highly appreciated and all of them have been taken into consideration, so changes have been applied as follow:

  1. Who are the authors of the manuscript?

Author response: The author is Dr. Ana Pérez Escoda. This data was omitted because of the blind review process.

Comment 1: In the Introduction, references to sources prove that social media as a significant promoter of engagement in society and industry must be included as part of formal education, as real scenarios in which citizens need to manage efficiently. New skills seem to be required to be taught at all levels of education to become exemplary practices for citizenship.

It has been proven that “This challenge requires a new curricular approach towards digital and media literacy that allows substantial progress in overcoming risks derived from media manipulation, inappropriate uses of the internet, inappropriate consumption of information, disinformation and infoxication and the spread of fake news”.

Moreover, considering the literature review, it is unclear what is proposed to be investigated.

Author response: The literature review has been modified in several paragraphs in order to clarify the framework, the purpose and make sense of the research questions presented. These changes can be found as follow:

  • Lines 36-40
  • Lines 52-57
  • Lines 79-84
  • Lines 88-97
  • Lines 103-105

Moreover, a new paragraph has been added focusing on the objectives of the study

Lines (126-131):

Having in mind the framework described the study purposed aims to analyze how information and fake news consumption affects the audience perception of media in terms of trust. For this purpose, there will be three specific objectives: firstly, to analyze sample use of social networks in terms of time, specific use and role adopted; secondly, fake news reception, and perception related to media outlets; and thirdly, study the perception of trust and its relationship with sample age.

Comment 2: The expert does not understand the scientific contribution that the authors make public. The introduction should also provide a deeper structure (1.1, 1.2, etc.) of the manuscript.

Author response: As suggested the author has introduced a deeper structure allowing a better understanding of the state of the art resulting: 1. Introduction; 1.1 Desinformation, fake news and social networks; 1.2 Media and audiences. In this regard more references has been added in order to offer a deeper study in the state of affairs.

  • de Oliveira, N.R.; Pisa, P.S.; Lopez, M.A.; de Medeiros, D.S.V.; Mattos, D.M.F. Identifying Fake News on Social Networks Based on Natural Language Processing: Trends and Challenges. Information 2021, 12, 38. https://doi.org/10.3390/info12010038
  • Somya Ranjan, S.; Brij B., G. Multiple features based approach for automatic fake news detection on social networks using deep learning. Applied Soft Computing 2021, 100, p. 106983. https://doi.org/10.1016/j.asoc.2020.106983
  • Sivasankari, S., Vadivu, G. Tracing the fake news propagation path using social network analysis. Soft Comput 2022, 26, 12883–12891. https://doi.org/10.1007/s00500-021-06043-2
  • Shrestha, A., Spezzano, F. Characterizing and predicting fake news spreaders in social networks. Int J Data Sci Anal 2022, 13, 385–398. https://doi.org/10.1007/s41060-021-00291-z
  • Williams, A.E.  Trust or Bust?: Questioning the Relationship Between Media Trust and News Attention. Journal of Broadcasting & Electronic Media 2012, 56(1), 116-131. https://doi.org/10.1080/08838151.2011.651186
  • Hanitzsch, T.; Van Dalen, A.; Steindl, N. Caught in the Nexus: A Comparative and Longitudinal Analysis of Public Trust in the Press. The International Journal of Press/Politics 2017, 23(1), https://doi.org/10.1177/1940161217740695
  • Tsafati, Y. Online News Exposure and Trust in the Mainstream Media: Exploring Possible Associations. American Behavioral Scientist 2010, 54(1), 22-42. https://doi.org/10.1177/0002764210376309
  • Ren, J.; Dong, H.; Popovic, A.; Sabnis, G.; Nickerson, J. Digital platforms in the news industry: how social media platforms impact traditional media news viewership. European Journal of Information Systems 2022, https://doi.org/10.1080/0960085X.2022.2103046
  1. Materials and Methods

The sentence (lines 157 and 158) is probably unfinished. In another case, a period should be added at the end of the sentence.

           Author response: Indeed, it was unfinished, now it is completed: “The quantitative research was based on an ethnographic perspective as the methodology aims at describing the qualities of a phenomenon, in this case the sample´s perception before answering questions related to social media consumption, fake news reception and perception and trust towards media”

General impression.

  1. According to the expert, the authors should clarify the goal and tasks of the research so that they explicitly emerge from the literature review as unsolved problems.

Author response: The aimand purpose of the study can be read in the Introduction (lines 41-47): “In this sense, the purpose of this work is to provide not only a descriptive picture but a deeper one on how information and fake news consumption affects the perception of media in terms of trust. Through an exploratory (which allowed a systematic approach to a reality) and ethno-graphic approach (since, based on this knowledge, actions were proposed), the possible strategies to be developed through the promotion of digital literacy in social networks are defined in order to counteract a situation that not only lasts over time, but seems to have come to stay”.

  1. It is necessary to specify the proposed approaches for achieving the goal and solving the tasks in Materials and Methods.

Author response: a new paragraph has been added focusing on the objectives of the study. Lines (126-131). The study presented is descriptive with mix-method quantitative and qualitative. It is not intended to solve any task, just drawing research objectives and giving answers to them as it is described in the Results.

  1. In the Discussion, the authors should compare the results with the known ones more clearly.

Author response: The references added has been integrated in the Discussion so results compared with known ones from previous studies can be better identified. The paragraph can be found lines 426-431.

  1. Conclusion section should be extended using: 1) numerical results obtained in the paper; 2) limitations of the proposed method; 3) perspectives of future research.

Author response: Following the reviewer suggestions limitations and future perspectives have been added to Conclusion paragraph (lines 460-466).

Reviewer 2 Report

The authors raise the topic related to the credibility of content on the Internet and the perception of it by users. A very interesting and timely topic. The authors present a proper introductory introduction to the topic. However, I miss the related works or state of art chapter. It has to be added. The references to the introduction itself are, in my opinion, insufficient.

The analysis was carried out on the basis of a questionnaire designed by the authors in order to answer the previously posed theses-questions. In my opinion, autorm has succeeded. They present their results, it is quite attractive to present and then comment on.

However, I have a note regarding tables and figures. All obtained values should be given with the same accuracy once adopted. It cannot be x.xx, then x.x or even x.

I believe that the article after these corrections - adding a chapter related to the reference to other works will be worth publishing.

Author Response

All suggestions are highly appreciated and all of them have been taken into consideration, so changes have been applied as follow:

Comment 1: The authors raise the topic related to the credibility of content on the Internet and the perception of it by users. A very interesting and timely topic. The authors present a proper introductory introduction to the topic. However, I miss the related works or state of art chapter. It has to be added. The references to the introduction itself are, in my opinion, insufficient.

Author Response: Following reviewer suggestion several studies have been added to the Introduction for a better background and framing in the state of art

  • de Oliveira, N.R.; Pisa, P.S.; Lopez, M.A.; de Medeiros, D.S.V.; Mattos, D.M.F. Identifying Fake News on Social Networks Based on Natural Language Processing: Trends and Challenges. Information 2021, 12, 38. https://doi.org/10.3390/info12010038
  • Somya Ranjan, S.; Brij B., G. Multiple features based approach for automatic fake news detection on social networks using deep learning. Applied Soft Computing 2021, 100, p. 106983. https://doi.org/10.1016/j.asoc.2020.106983
  • Sivasankari, S., Vadivu, G. Tracing the fake news propagation path using social network analysis. Soft Comput 2022, 26, 12883–12891. https://doi.org/10.1007/s00500-021-06043-2
  • Shrestha, A., Spezzano, F. Characterizing and predicting fake news spreaders in social networks. Int J Data Sci Anal 2022, 13, 385–398. https://doi.org/10.1007/s41060-021-00291-z
  • Williams, A.E.  Trust or Bust?: Questioning the Relationship Between Media Trust and News Attention. Journal of Broadcasting & Electronic Media 2012, 56(1), 116-131. https://doi.org/10.1080/08838151.2011.651186
  • Hanitzsch, T.; Van Dalen, A.; Steindl, N. Caught in the Nexus: A Comparative and Longitudinal Analysis of Public Trust in the Press. The International Journal of Press/Politics 2017, 23(1), https://doi.org/10.1177/1940161217740695
  • Tsafati, Y. Online News Exposure and Trust in the Mainstream Media: Exploring Possible Associations. American Behavioral Scientist 2010, 54(1), 22-42. https://doi.org/10.1177/0002764210376309
  • Ren, J.; Dong, H.; Popovic, A.; Sabnis, G.; Nickerson, J. Digital platforms in the news industry: how social media platforms impact traditional media news viewership. European Journal of Information Systems 2022, https://doi.org/10.1080/0960085X.2022.2103046

The analysis was carried out on the basis of a questionnaire designed by the authors in order to answer the previously posed theses-questions. In my opinion, autorm has succeeded. They present their results, it is quite attractive to present and then comment on. However, I have a note regarding tables and figures. All obtained values should be given with the same accuracy once adopted. It cannot be x.xx, then x.x or even x.

            Author response: All data from figures and tables has been corrected with the same accuracy adopted.

Round 2

Reviewer 1 Report

All comments are more or less taken into account

Reviewer 2 Report

The authors responded to my comments. Thank you